# Language Acquisition, Neutral Change, and Diachronic Trends in Noun Classifiers

## Abstract

Languages around the world employ classifier systems as a method of semantic organization and categorization. These systems are rife with variability, violability, and amiguity, and are prone to constant change over time. We explicitly model change in classifier systems as the population-level outcome of child language acquisition over time in order to shed light on the factors that drive change to classifier systems. Our research consists of two parts: a contrastive corpus study of Cantonese and Mandarin child-directed speech to determine the role that ambiguity and homophony avoidance may play in classifier learning and change followed by a series of population-level learning simulations of an abstract classifier system. We find that acquisition without reference to ambiguity avoidance is sufficient to drive broad trends in classifier change and suggest an additional role for adults and discourse factors in classifier death.

## 1 Introduction

Classifier and measure word systems are common across the world's languages. While they are the most common and most associated with Southeast and East Asia, they are also present in some languages of South Asia, Australia, the Pacific, and the Americas among others (Aikhenvald, 2000). While systems vary language-to-language, they share some general properties. They divide up the space of nouns along some semantic space, often encoding encode lexical semantic information including animacy, concretely, and size and shape categories. For example, Mandarin has classifiers for long objects (e.g., *tiáo* 條), some animals (*zhī* 隻), and vehicles (*liàng* 輛). On the other hand, some classifiers do not seem to pick out anything in particular, like the Mandarin general classifier *gè* 個 or instead pick out extremely narrow, almost lexicalized classes, like *zūn* 尊 which as a classifier applies only to certain colossal metal objects like cannons Buddhist statues (Gao and Malt, 2009).

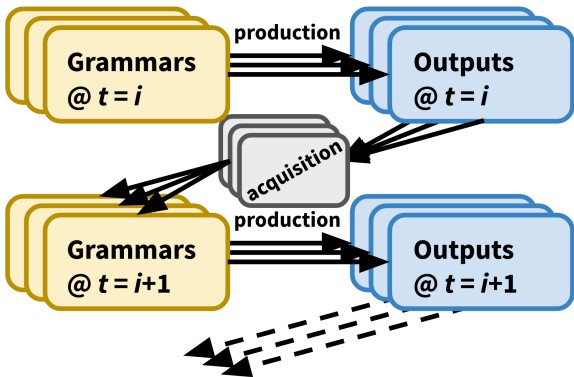

Figure 1: The Z-model of change extended to a population setting

Compared to most inflectional noun class systems, classifiers are more subject to variable discourse conditions. Several classifiers may be used grammatically with a given noun as conditions allow. For example, 'a goat' may be expressed with the animal classifier *zhī* or general classifier *gè*, but also *tiáo* or *tóu* 頭 used for livestock (Erbaugh, 1986). The balance of semantic specificity, arbitrariness, and variability presents a challenge for native learners. How do individuals acquire both the semantic conditions and arbitrary lexical patterns of classifier systems?

Parallel to this, classifier systems are subject to constant change, both for language-internal reasons (e.g., grammaticalization of new classifiers, word death of old classifiers) and external ones, particularly contact (Aikhenvald, 2000). In Mandarin Chinese, Erbaugh (1986) illustrates a few cases of changes in classifier usage over the past 3500 years. *Gè* 個, the overwhelming majority catch-all classifier in modern Mandarin only gained this status during the during the Qing Dynasty (CE 1644-1912). Since the Tang dynasty a millenium prior, *méi* 枚 was the default, but it has since been relegated to a niche classifier for small needle and badge-like objects alone. Both *gè* and *méi* began as niche classifiers in their respective eras before grad-

ually generalizing. In a similar vein, Habibi et al. (2020) explore how linguistic categories change through chaining, via the usage of Mandarin Chinese classifiers in the past half century. The latter two studies discuss the development of Mandarin classifiers over time. They are based on careful research, but they are also limited to a single language and Erbaugh (1986) in particular stops short of a quantitative assessment.

We provide a computational analysis of diachronic trends in classifier systems which complements prior developmental and historical research. We approach the problem in two ways. First, we provide a quantitative analysis of classifiers in Cantonese and Mandarin child-directed speech to investigate the possibility of a functional role for classifiers as disambiguators which could influence the direction of child-driven change. Second, we model a simulated classifier system using a population-level transmission model to determine how language acquisition may drive trends in classifier patterns over time. We find support for input sparsity and learning, without reference to specific functional concerns, as a primary driver for gradual classifier generalization over time.

## 1.1 Outline

The paper is organized as follows. Section 2 surveys cross-linguistic patterns in classifier acquisition and summarizes work connecting language acquisition to change. Section 3 is a comparative study of adult classifier use in Cantonese and Mandarin child-directed speech corpora. This motivates our simulation. We show that the historical development of classifiers is unlikely to be driven by functional communicative concerns such as ambiguity avoidance on behalf of the learner. Section 4 describes are simulation, which falls under the umbrella of *neutral* or *drift*-based models of change. We find that classifiers tend to generalize, fail to maintain distinct semantic features, and also cannot go out of use randomly. Section 5 discusses the implications of our simulation in reference to Chinese in particular and provides suggestions for future extensions to this line of work.

## 2 Classifier Learning and Change

Language acquisition has long been implicated as a driver of language change (Paul, 1880; Halle, 1962; Andersen, 1973; Baron, 1977; Lightfoot, 1979; Niyogi and Berwick, 1997; Yang, 2002; Kroch, 2005; van Gelderen, 2011; Yang, 2016; Cournane, 2017; Kodner, 2020, *i.a.*), and this is particularly been true for morphology, where child over-productivity errors (Marcus et al., 1992; Mayol, 2007) quite often mirror the processes of analogical change, which itself closely connected to productivity (Hock, 2003, p.446).

Classifier systems are not structurally morphological and do not trigger syntactic agreement like inflectional noun class systems, but they share some key properties, both in their use and acquisition. Both often encode lexical semantic information including animacy, concretely, and size and shape categories. For example, the Bantu language Shona has noun classses for mostly long-skinny things (e.g., class 11 *ru-*), classes for animals (e.g., class 9 *(i)-*), and miscellaneous classes (e.g., class 7 *chi-*) which correspond broadly to the Mandarin classifiers described in Section 1. Both noun classes and classifiers may be semantically porous with many lexical exceptions. And while classifiers are generally more variable than inflectional classes, the later may also show variability. In Shona again, people usually take the class 1 *mu-* prefix (*mu-nhu* 'person'), but if a speaker wishes to highlight that a person is particularly tall and thin, they may employ the long-skinny class 11 prefix (*ru*-nhu).

Learners of classifier languages exhibit generally competent classifier use by age 4 or 5, though they show some command over their syntax much earlier (Chien et al., 2003; Tse et al., 2007; Liu, 2008). Children are prone to overusing the general or default classifier in Japanese (Uchida and Imai, 1999), Mandarin (Liu, 2008), Cantonese (Tse et al., 2007), and Vietnamese (Tran, 2011), similar to the over-extension of default patterns in morphology (Pinker and Prince, 1994). They take longer to acquire rare classifiers and those with complex semantic restrictions (Yamamoto and Keil, 2000).

A division of classifiers into semantically well-defined and arbitrary is well-motivated by a series of experiments carried out by Gao and Malt (2009) on Mandarin. This further clarifies what the learning task entails. Children must work out whether classifiers are lexically defined or generally apply to a given semantic class. This is consistent with observed developmental trajectories: learners pass through an early lexicalized stage in which classifiers are defined narrowly by which lexical items they match with rather than their general semantics, followed by a higher than adult-rate use of generic

classifiers, before settling on an adult-like distribution (Erbaugh, 1986). This is parallel to the classic inflectional learning trajectory, a pre-generalization period, followed by over-generalization of defaults, followed by settling on an adult-like distribution.

Erbaugh (1986) explicitly connect classifier acquisition to change in Chinese and notes several parallels between Chinese classifier acquisition and change. Most relevant for the present study, classifiers are narrowly, perhaps lexically, defined when they enter the language and then trend towards generality, and they apply to concrete objects with real-world identifiable semantics before abstract concepts, in line with children's preference for real world referents in their dialogues.

Taken together, classifier systems have enough in common with inflectional class systems that their acquisition and change can be modeled similarly. Linguistic transmission, the passing of a language from one generation to the next through native language acquisition (Weinreich et al., 1968), provides a fundamental role for acquisition in change. Andersen (1973) formalizes change as the long-term consequence of abductive processes in language acquisition through his Z-model: Speakers have some internal grammar which generates a set of linguistic examples which serve as the input to the next generation. The next generation acquires a grammar based on these finite inputs and produces outputs for the next generation. This process proceeds indefinitely. Abudction is error-prone, and differences between the grammars of the first and second generation are tantamount to change.

But language change is fundamentally a population-level process (Weinreich et al., 1968; Labov, 2001), so the Z-model must be thought of as countless parallel lines of transmission and not a single Z-shape. Additionally, transmission does not proceed through discrete generations, but rather is continuous across age cohorts in the population. Children learn from their parents, but begin to orient towards their peers and community as toddlers (Roberts, 1994), eventually culminating in extremely complex social networks in their teen years (Eckert, 1989), so the Z-model should be staggered both across the population and across time. This view, diagrammed in Figure 1, forms the conceptual basis of our simulation.

A population-based transmission model in which what is acquired is driven primarily by the input and not additional functional factors may be described as *neutral*. This is often assumed as the baseline in biological evolution (Neutral Theory; Kimura, 1983), and may be for language change as well (Kauhanen, 2017). The following section tests an alternative, that classifiers emerge to decrease homophony, before adopting a neutral approach.

## 3   Classifiers and Homophony

This section quantifies classifier use in Mandarin and Cantonese child-directed speech (CDS). Their systems are quite similar, both having descended from Middle Chinese. Since their divergence, the languages have undergone substantial phonological divergence resulting in much less syllable diversity in Mandarin.[1] For this reason, Mandarin is expected to show more homophony than Cantonese, though this is offset by an increase in polysyllabic words in Mandarin.

Disambiguation of homophones is one possible function of classifiers and a potential functional (i.e., non-neutral) driver of change. More elaborate classifier systems may develop in response to more rampant homophony. We compare Mandarin and Cantonese CDS to determine whether homophony avoidance is plausibly part of the child's role in the development of the Chinese classifier systems. If true, we would expect Mandarin CDS to show more noun form ambiguity than Cantonese *and* show more classifier disambiguation of homophonous word types.

All POS-tagged Mandarin and Cantonese corpora were extracted from the R conversion (Sanchez et al., 2019) of the CHILDES database of child-directed speech corpora (MacWhinney, 2000) except for Erbaugh, which could not be retrieved. The first two data rows of Table 1 summarizes the corpora, and (1)-(2) provide example utterances. We extracted classifiers tagged cl from adult speech in the corpora if they preceded a noun, or preceded an adjective or adverb which preceded a noun, along with the noun itself. Sometimes transcription lines did not align with the characters, which we attempted to resolve by tracking known classifier characters and examining the neighbourhood of the incongruency in the sentence. A handful of cases could not be resolved, so they were omitted. We omitted classifier pro-forms since no noun surfaces in the utterance. We define homophones as two word forms with different characters

---

[1]e.g., Mandarin's 4 (5) tones, and ∼34 syllable rimes compared to Cantonese's 9 and 60.

| Corpus | #Types (%Poly) | %Types HP | %Disamb | #Toks (%Poly) | %Toks HP | %Disamb | #Cl |
|---|---|---|---|---|---|---|---|
| Cantonese | 1182 (55.6) | 4.653 | 20.000 | 19880 (21.4) | 7.706 | 6.201 | 76 |
| Mandarin | 2151 (71.8) | 7.345 | 22.785 | 30891 (41.8) | 28.558 | 6.506 | 149 |
| Mandarin$_{type}$ | 1182.2 (63.0) | 8.815 | 20.430 | 28066 (39.0) | 28.264 | 6.776 | 140.0 |
| Mandarin$_{tok}$ | 221.9 (43.0) | 4.778 | 16.981 | 19880 (31.9) | 23.431 | 3.078 | 98.5 |

Table 1: Cantonese, Mandarin, avg. type freq-controlled Mandarin$_{type}$, and avg. token freq-controlled Mandarin$_{tok}$ corpus size, %nouns polysyllabic, % nouns which are homophonous (HP), the % of homophonous nouns which are disambiguated by their classifiers, and # classifiers.

but the same transcription.

(1) **Cantonese** (HKU-70; Fletcher et al., 1996)
INV: 你 得 一 個 啤啤 zaa4 .

```
%mor:  pro|nei5=you stprt|dak1
num|jat1=one cl|go3=cl
n|bi4&DIM=baby sfp|zaa4 .
```

(2) **Mandarin** (Zhou1; Zhou, 2001)
MOT: 画 个 小 圆圈 宝贝 .

```
%mor: v|hua4=draw
cl|ge4 adj|xiao3=small
n|yuan2quan1=circle
n|bao3bei4=treasure .
```

Since corpus size could have a substantial effect on the ratios reported in the corpora, we opted to downsample the Mandarin corpus to match the size of Cantonese and compare both the downsampled and raw Mandarin. We dropped out Mandarin tokens selected uniformly at random until the the corpus matched the Cantonese corpus in type or token count. This was repeated for 100 trials and averaged. The resulting Mandarin$_{type}$ and Mandarin$_{tok}$ are the last two rows in Table 1. When matched for types, the Mandarin corpus has substantially more polysyllabic words than Cantonese, and when matched for tokens, it has substantially more polysemous tokens. It also has a wider range of classifiers and measure words. The increase in polysyllabicity in Chinese varieties is traditionally taken to be a response to increased homophony due to phonemic mergers (Karlgren, 1949).

The table also shows the rates of homophonous word types in the corpora as well as the proportion of those which are *disambiguated*. We defined a homophonous word type as disambiguated if every homophone is attested with at least one classifier not attested with any other homophone, and a disambiguated word token as any token which belongs to a disambiguated word type. Despite the increase in polysyllabicity, Mandarin is still much more ambiguous than Cantonese. Nevertheless, its homophones are not significantly more disambiguated.[2]

This analysis is consistent with (but does not prove) the idea that polysyllabicity emerged in Chinese in a response to ambiguity. In contrast, it does not support a role for homophony avoidance as a motivation for the classifier system. Even though the Mandarin acquisition corpora attest more classifiers and measure words, only about 1/5 of homophonous types and 1/18 of homophonous tokens are disambiguated by classifiers. The fact that tokens are much less likely than types to be disambiguated, and that the type disambiguation rate declines as the number of types fall in Table 1, also indicates the type disambiguation rate is generous and inflated by low frequency and edge cases. Additionally, Mandarin does not exhibit more classifier disambiguation even though it is more homophonous than Cantonese. Given this, we can justify our major modeling assumption, that changes to the classifier system are not primarily driven by communicative concerns.

## 4 A Classifier System in a Population

The empirical analysis in the previous section motivates a neutral model of change (Kauhanen, 2017) for the Chinese classifier system. In this section, we introduce a population-level model of linguistic transmission to investigate the dynamics of classifier systems over time. We describe the details of our simulation, including the algorithm and parameters, their relevance, and their specific empirical motivations. We then discuss our findings across different parameter settings, and consider their implications in the study of classifiers, learning, and language change.

### 4.1 Methodology

At a high level, our simulation consists of a population of entities sorted by age into "children" who are still acquiring a classifier system and "adults"

---

[2]One-sided Z-test on Cantonese vs. Mandarin$_{type}$ types is insignificant: $Z = 1.570$ at $\alpha = 0.05$, while test on Cantonese vs. Mandarin$_{type}$ tokens shows that Cantonese has significantly *fewer* disambiguated homophones $Z = -2886.511$.

with productive representations of classifiers. At the start of each iteration, the oldest adult "dies," a new child is "born," and every entity's age is incremented, with the eldest child maturing into an adult, as we describe later. During the iteration, adults interact with a subset of children, and children learn from these interactions. Crucially, transmission flows from the pool of adults as a whole. Ages are continuous, and children can learn from the youngest adults as well as the oldest. This admits the diffusion of innovations, thus actuating the change (Labov et al., 1972) and potentially yields significant variable input for the learners. Algorithm 1 formalizes the population model.[3]

---

**Algorithm 1** Simulation iteration algorithm

---

1: $CH \leftarrow$ List of children of size $K$
2: $AD \leftarrow$ List of adults of size $N - K$
3: **for** $s := 1...S$ **do**
4:      Delete $AD[-1]$ as oldest adult "dies"
5:      Move $CH[-1]$ to $AD[0]$ as oldest child "matures" using productivity method PROD
6:      A new child is "born" at $CH[0]$
7:      **for all** $adult \in AD$ **do**
8:          $mutate\_classifier\_set(adult, A, D)$
9:          **for** $i := 1...I$ **do**
10:             $child \leftarrow$ random child $\in CH$
11:             $nouns \leftarrow J$ random lexical items
12:             $interact(adult, child, nouns)$
13:          **end for**
14:      **end for**
15: **end for**

---

Classifiers in the simulation are represented as abstract binary semantic features (abstract, but conceptually equivalent to $\pm$ANIMATE, $\pm$FLAT, etc.). These are encoded as binary vectors of size $F$. Lexical items are organized along a Zipfian distribution, since it is observed to fit token frequencies well across languages (Zipf, 1949; Baayen, 2001; Yang, 2013). At initialization, each adult has the same set of $C$ classifiers. This set includes at least one "most general" classifier, while other classifiers are initialized randomly. Children are initialized so that at the first iteration it is as if the eldest child has gone through $K$ iterations (and therefore rounds of interactions) already.

Most simulations run using a feature hierarchy: features are organized hierarchically with one most generic parent feature and up to $B$ sub-features such that there are $F$ total features. The presence of a sub-feature implies the presence of its parent features. Depending on the simulation, up to $H$ features are assigned in this manner. A flat repre-

sentation would make for ambiguous results since it would be unclear whether more features correspond to a more general or more specific classifier.

Children learn as follows: In each iteration, children observe many classifier-noun pairs. They add the features on the noun to a running tally of observed features for the classifier, but crucially, they do not yet know which features actually select the classifier, since nouns may contain properties that are just incidental and unrelated to the particular choice of classifier. After some $K$ iterations, a child matures. The child evaluates whether a classifier productively expresses a feature by comparing its observations against a threshold for productivity provided by the Tolerance Principle (TP; Yang, 2016), a quantitative model of productivity learning which has been successful in accounting for developmental patterns in morphology and elsewhere.

For a given feature $f$ observed with a noun paired with the classifier $c$, if the number of attested paired noun types that *do not* express that feature (the exceptions, $e_f^c$) is less that the tolerance threshold $\theta_f^c$ for that classifier, then that feature will be productive on the classifier. The tolerance threshold is calculated as in Eqn. 1. $N^c$ is the total number of noun types attested with the classifier.[4]

$$e_f^c < \theta_f^c,$$
$$\text{where } \theta_f^c = \frac{N^c}{\ln N^c} \quad (1)$$

We provide a role for adults as drivers of change by introducing two additional parameters. An adult may drop a classifier with probability $D$ by setting it to be non-productive on all features, and provided there is an opening (ie. some classifier is non-productive on all features) add a new classifier with probability $A$. This is taken to represent choices available adults in response to discourse and sociolinguistic factors. We believe that such factors affecting adults may be responsible for the death of high frequency general classifiers, since no child in a neutral model of change would fail to learn something so well and so diversely attested.

There is always a worry that a highly parameterized simulation will do something akin to overfitting the pattern the researcher is trying to recreate. To guard against this, we test a wide range of parameter settings to confirm that the system's dynamics are inherent to the model and not driven

---

[3]Parameterized according to Table 2 in the Appendix.

[4]See (Yang, 2018) for a summary of the TP's psychological motivation and mathematical derivation.

by a convenient parameterization. To the extent possible, default parameters were motivated empirically (e.g., Zipfian token frequency distribution) or according to practical concerns (e.g., if the number of classifiers far exceeds the number of semantic features $C \gg F$, most classifiers will be synonymous and redundant). A full list of parameters available to the model are presented in Table 2 in the Appendix.

We ran four sets of simulations testing distinct hypotheses. The first set included 58 simulations, and did broad sweep of the parameter space, testing parameter values on either side of their defaults as well as different non-numeric parameters. The second set included 37 simulations, and varied the probability that adults add or drop classifiers, since these values are internal to the simulation. The third set included 20 simulations, running 4 parameter settings in repetition 5 times to weed out uniquely random outcomes. Finally, the fourth set included 15 simulations, varying a few parameters but running and repeating settings for 5,000 iterations to observe what happens in the very long term. In total, we ran and examined 130 simulations.[5]

### 4.2 Results

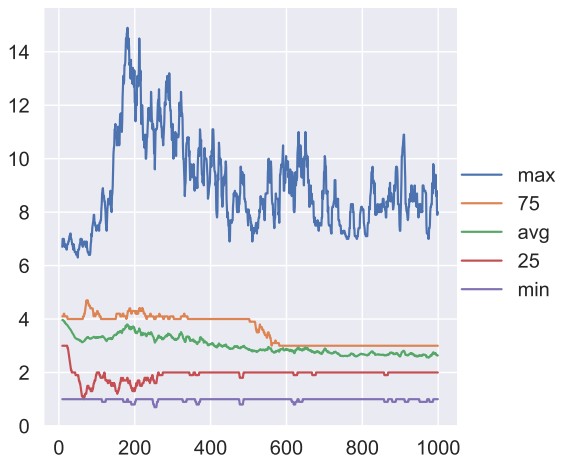

Figure 2: Typical outcome for a simulation run on default parameters

We found that many parameterizations admitted complex dynamics, and successive runs with the same settings sometimes yielded different outcomes. Nevertheless, particular trends emerged. We observe three findings repeated across a range of settings which we believe characterize neutral

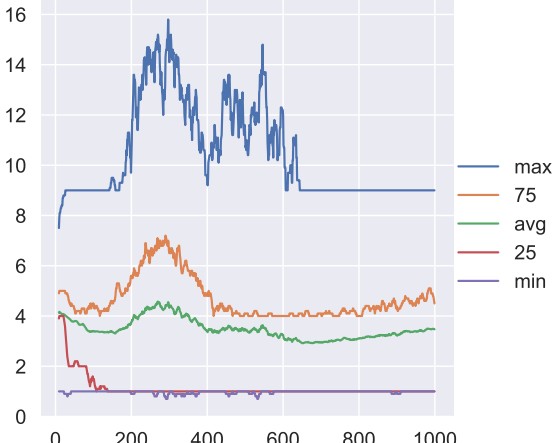

Figure 3: Atypical outcome on default parameters: mean # features trends up

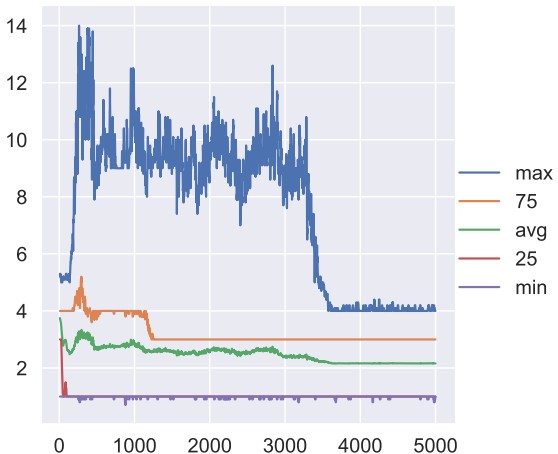

Figure 4: Simulation run for 5,000 iterations, default parameters.

transmission of classifiers more broadly. Figures 2-8 demonstrate these trends.[6] They show how the maximum, minimum, and average number of features, as well as the 25th and 75th percentiles, averaged over the 10 youngest adults, change over time.

Figures 2 and 3 show the behavior of the simulation with default parameters. parameters described in the appendix. We chose these settings as the simplest that still admit interesting dynamics into the system. The average number of features per classifier trends downwards but does not do so monotonically. This is consistent with the diachronic trend observed by Erbaugh (1986) in which general classifiers emerge from more specific classifiers over time. This does not, however, predict that classifier systems should devolve completely, either in

---

[5]All code, including the specifications of our sets of simulations, is publicly available at `url.omitted.for.review`

[6]Parameterizations specified in Table 4 in the Appendix.

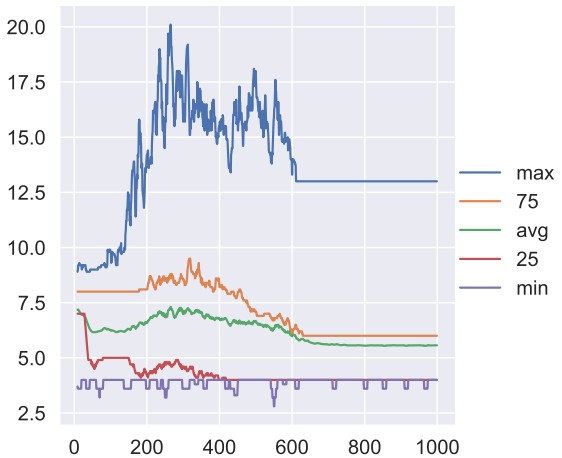

Figure 5: A simulation variable branching feature hierarchies

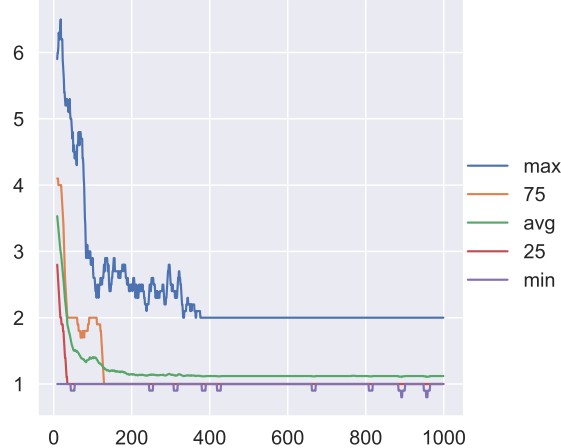

Figure 7: A simulation with random classifier dropping showing rapid contraction

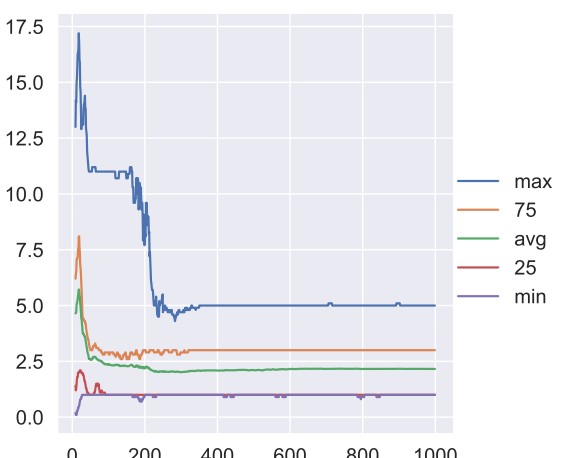

Figure 6: A simulation with multiple feature initialization showing rapid contraction

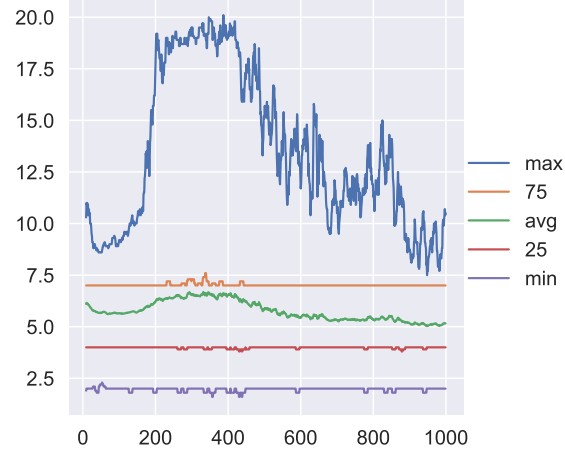

Figure 8: Simulation with variable feature initialization and 10x new classifier adding

our simulations or in the diachronic data. Our simulations often settle on a steady state after many iterations (Fig. 4) and may occasionally reverse direction after a period of near-stability (Fig. 3).

The steady state of the simulations could indicate insufficient churn in the set of available classifiers. To test this, we increased the rate of adults adding classifiers by a factor of 10. This did not have a significant effect on the average (Fig. 8), and failed to consistently stave off the slow gradual generalization seen in earlier simulations. Robustness to this parameter choice further confirms that learning, and not adult innovation, to combat ambiguity, for example, is driving the trends observed here.

Finally, if new classifiers were initialized with a random, potentially large, number of features (Fig. 6), or if adults drop random classifiers instead of the most general ones (Fig. 7), the system rapidly and consistently reduces to one with a few more general classifiers. This makes sense, since a new classifier that is very semantically restricted is unlikely to be sufficiently attested for children to learn all of its features. Similarly, if classifiers are dropped randomly, highly specific classifiers will be dropped with some probability. Children will have less evidence to learn them, and they will not be acquired in their full specificity.

## 5 Discussion and Conclusion

In this paper, we advocate a view of language change as a natural outcome of language acquisition over time and across a population. This acquisition-driven view of change provides insight into the long-term dynamics of classifier systems through a cross-linguistic corpus study of modern Chinese child-directed speech and a population-

level simulation of classifier change.

The cross-linguistic study (Section 3) contrasts Mandarin and Cantonese, two closely related but not mutually intelligible languages with a recent common ancestor, to test the hypothesis that classifier use is driven by homophony avoidance. We found that though Mandarin child-directed speech has substantially more homophonous types than Cantonese, its classifiers actually disambiguate homophones significantly less often. This is contrasted with polysyllabicity in Mandarin, which does show a trend consistent with homophony avoidance.

This result motivates a neutral model of classifier change driven by matters of learning and input sparsity not primarily concerned with functional pressures. We apply the Tolerance Principle (TP), a model of productivity learning, to our population-level simulation and observe general trends. The TP was chosen because it successfully models U-shaped learning trajectories in morphology where learners develop through memorization to over-generalizing phases. This is similar to the developmental pattern observed in classifier learning. Children begin by memorizing classifiers and the nouns they apply to, then move to over-use of general classifiers. A similar trend towards generalization is observed empirically in the history of Chinese classifiers. New classifiers are specific when they are introduced and tend towards generality over time. This is not a lockstep relationship along the lines of "ontogeny recapitulates phylogeny," but two parallel trends which emerge independently from the same learning process. Our population-level simulation of TP learners (Section 4) achieves this pattern under a wide range of parameter settings, providing support for the role of learning and neutral processes in this change.

### 5.1 Future Work

One question that has yet to be resolved is what could have caused the replacement of the Tang-Qing general classifier *méi* with the Qing-modern *gè*. We believe that the solution likely lies in discourse factors. Adults may choose more specific classifiers over the most general one in order to emphasize qualities of the noun being modified. This would explain why *méi* was not completely replaced when it lost its generic status and was instead reduced to a narrow semantic scope. Change here may be modeled as a sociolinguistic variable

(Labov, 1994). However, such socially conditioned change is lead by young adults rather than young learners. A fully developed mechanism for changes in the classifier system would require modeling both acquisition-driven and sociolinguistic change simultaneously.

As an initial test of this hypothesis, we compared simulations in which adults drop the most generic classifier with some low probability (representing a sociolinguistic choice to prefer an innovative classifier) against simulations in which adults drop classifiers at random. We find that the former allows for the expected slow generalization of classifiers while the latter causes the system to rapidly collapse. We interpret this as supportive of the discourse driven account, but sophisticated extensions would be needed to demonstrate it. Similarly, the population model could be extended to better capture sociolinguistic network topology (Milroy and Milroy, 1985; Kodner and Cerezo Falco, 2018).

Parallel to this, a complete account would incorporate more concrete semantic representations and algorithms to represent word coining into our simulations (Habibi et al., 2020; Xu and Xu, 2021). Our simulation does not account for the creation of new classifiers, which tend to emerge through grammaticalization of nouns (Aikhenvald, 2000), nor does it provide a structured means for representing classifier semantics beyond the abstract hierarchies which we employed. Semantic chaining (Ramiro et al., 2018; Xu and Xu, 2021) is a promising candidate approach. Our population-level acquisition-driven approach provides a base upon which to develop fully featured diachronic models of classifier systems.

### 5.2 Conclusion

This paper adopts a view of language change that is primarily driven by children acquiring their native languages with additional changes led by adults. This dual perspective provides a place for both grammar learning and sociolinguistic discourse factors as mechanisms for change. Classifier systems are a natural juncture for these two types of change since they are both deeply embedded in the grammar and show heavy optionality, variablity, and discourse sensitivity. We further clarify in our empirical study of Cantonese and Mandarin as well as the simulations that child-driven change to classifier systems is neutral with respect to function.

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

## A   Appendix

| Parameter | Value | Explanation |
|---|---|---|
| $S$ | 1000 | No. simulation iterations |
| $N$ | 200 | No. total individuals |
| $K$ | 40 | No. children |
| $V$ | 1000 | No. nouns in lexicon |
| $C$ | 25 | No. classifiers in lexicon |
| $F$ | 50 | No. features |
| $G$ | 4 | Max no. noun features |
| $H$ | 3 | Max no. init classifier features |
| $B$ | 3 | Max branching factor within a feature hierarchy |
| $I$ | 5 | No. interactions by adults |
| $J$ | 5 | No. lexical items per interaction |
| $A$ | 0.01 | Prob. add classifier per iteration |
| $D$ | 0.01 | Prob. drop classifier per iteration |
| PROD | TP | Method for productivity in acquisition |
| LEX_TYPE | Zipf | Distribution type of nouns in the lexicon |
| CLASS_INIT | hierarchy, single | Method for classifier initialization, including feature hierarchy |
| FEAT_INIT | fixed | Method for initializing a feature hierarchy, dependent on $B$ |
| CLASS_DROP | general | Target for dropping classifiers |

Table 2: A list of simulation parameters, their default values, and what they do. Non-numeric parameters are further described in Table 3.

| Parameter | Value | Explanation |
|---|---|---|
| PROD | TP | Tolerance Principle (Yang, 2016) |
| | majority | Simple majority |
| LEX_TYPE | Zipf | Lexical items follow a Zipfian distribution (Zipf, 1949; Lignos and Yang, 2018) |
| | uniform | Lexical items follow a uniform distribution |
| CLASS_INIT | identity | Classifiers are initialized as though an identity matrix |
| | random | Classifiers are initialized randomly with $H$ features |
| | hierarchy, single | Classifiers are initialized with 1 feature using a feature hierarchy |
| | hierarchy, multiple | Classifiers are initialized with 1 to $H$ features using a feature hierarchy |
| FEAT_INIT | fixed | Each feature in the hierarchy has $B$ children |
| | variable | Each feature in the hierarchy has 1 to $B$ children |
| CLASS_DROP | general | The classifier with the least number of features is dropped |
| | random | A random classifier is dropped |

Table 3: A list of possible arguments for each of the non-numeric parameters in our simulation. Explanations for each of parameter's purpose are found in Table 2 and in Section 4.1.

| Figure no. | Parameters |
|---|---|
| 2 | (used default) |
| 3 | (used default) |
| 4 | $S = 5000$ |
| 5 | FEAT_INIT = variable |
| 6 | CLASS_INIT = hierarchy, multiple |
| 7 | CLASS_DROP = random |
| 8 | $A = 0.1$, FEAT_INIT = variable |

Table 4: The parameters that the simulation presented in each figure ran on, where they differ from the default arguments listed in Table 2.

