# OpenReview forum: "Language Acquisition, Neutral Change, and Diachronic Trends in Noun Classifiers"
_aclweb.org/ACL/2022/Workshop/CMCL — Submitted to CMCL 2022_

### Official Review · Reviewer_t543 · 2022-03-26
**An interesting study on modeling the acquisition of classifier systems, though not so much on modeling language change.**

**Rating:** 6
**Confidence:** 4

**Review:**

This paper shows two sets of experiments in an attempt to explicitly model change in classifier systems as some outcome of child language acquisition. The topic is interesting, and the paper includes several interesting ideas, and has great suggestions about future work. However, there are some parts that are not clear to me. While it assumes that language acquisition may drive trends in classifier patterns over time, what the changes are have not been clearly defined. For example, in the introduction, the authors mentioned the replacement of the general classifier 'mei' with the currently more dominantly-used 'ge' over time; while this indeed is an interesting point of inquiry, the studies presented in the paper were not able to address this type of change. Alos, is adults' using more specific classifiers while dropping the general classifiers considered "a change"? If yes, this point should be elaborated. The authors mentioned that such selections may be related to discourses, or due to sociolinguistic factors, but the experiment results only show that children tend to acquire the (old or new) uses of classifiers presented by adults, and then stabilize the use of them, and therefore although I agree with the authors that the results support the trends of learning and neutral processes, they do not seem to be related to language change.

Below are some minor suggestions about the organization and the contents.
1. Introduction: Need to reconsider what role the example of 'mei'-'ge' plays in the paper.
2. Section 2 can directly highlight how similar classifier systems and inflectional agreement systems are in terms of modeling, and then elaborate how the classifier system can be modeled. Currently, the first half of the section suggests the otherwise (i.e., they are very different grammatical properties).
3. The differences between classifiers and measure words do not seem to be relevant to the current study. Maybe using classifiers as a cover term would be enough.
4. Mandarin 'ge' although is a general classifier, it functions as an individualizer (i.e., its function is to pick out one individual entity in particular); the description of it at the beginning of the paper is a bit misleading.

---

### Official Review · Reviewer_8g4d · 2022-03-27
**Analyzing and simulating the acquisition of noun classifiers over generations**

**Rating:** 6
**Confidence:** 3

**Review:**

This paper analyzes classifier systems that categorize nouns in Mandarin and Cantonese child-directed speech and in light of the findings of this analysis, proposes a population-based model of classifier acquisition and change over time. Models with various setups produce classifiers that are more generalized with fewer features as the simulation progresses, seeming to indicate a neutral learning scheme based on inputs rather than functional factors.

**Pros**

The authors explore various setups of simulations and an interesting set of results are reported, providing insightful information regarding possible theories of the evolution of classifiers.

The authors motivate their population-based transmission simulation with findings from the analysis of actual human data and also relevant developmental literature, focusing on neutral learning from input.

The simulation results point to an inclination toward the production of generalized classifiers, seeming to confirm the authors’ motivation behind the implementation of these models.

**Cons**

I was confused by the significance findings reported in footnote 2. Even though for the types, the difference is insignificant, for the tokens, Mandarin has more disambiguated homophones. Then, in Section 5, it is written that Mandarin classifiers disambiguate homophones significantly less often, which I thought was more in line with what was reported in Table 1. It would be nice if the authors clarify what numbers were exactly being contrasted in these tests to provide a stronger motivation for their choice of simulation model.

The simulation models seem to be rather independent of the child-directed speech data, apart from the idea of neutral learning. It would be informative to contrast these models with a model that follows functional objectives or models that include noun attributes and classifier features akin to the ones observed in real human data.

**Writing**

004 amiguity -> ambiguity

031 Remove 'encode'

032 concretely -> concreteness (?)

063 'during the' repeated

105 are -> our (?)

119 is -> has

130 concretely -> concreteness (?)

286 'the' repeated

604 variability

---

### Official Review · Reviewer_eXxh · 2022-03-28
**Hard to identify the main messages the authors want to convey**

**Rating:** 4
**Confidence:** 4

**Review:**

There's something potentially of interest here but this version of the paper feels very disjointed. It is hard to identify the key thread of ideas that the authors intend to be the main take-home message, and it is hard to understand the logical links between various sections of the paper.

Perhaps the key concrete point that I can identify is that classifiers tend to change from narrowly-applicable to more general over historical time, and the model presented in section 4 correctly recreates this effect. This is the most meaningful common thread I can find running through the paper. The empirical point about the historical change is buried in the middle of section 2 (lines 174-183) in amongst discussion of all sorts of other things, and the fact that the simulations recreate this pattern is mentioned (lines 468-473) as one of three findings in section 4.2. (Even this would be clearer if the authors explicitly mentioned that more features means more specific. And at line 460, it should be stated explicitly that the graphs show the average number of features *per classifier*.) In between, there's very little to help the reader connect the dots: for example, the intro to section 4 says nothing about what kind of patterns the authors are hoping to recreate (only that they will "investigate the dynamics of classifier systems over time").

The logic of the relationship between section 3 and section 4 seems muddled. In section 3 itself, the corpus study is presented as evidence *against* the non-neutral, ambiguity-avoidance hypothesis, and this is presented as justification for adopting a model without functional pressures in section 4. But the abstract, for example, says that "acquisition without reference to ambiguity avoidance is *sufficient* to drive broad trends", which makes it sound like the paper will be treating the neutral hypothesis as the simpler/null hypothesis and showing that additional assumptions (i.e. non-neutrality) are unnecessary. So there's a confusing inconsistency in the way the authors talk about which hypothesis (neutral or non-neutral) is the one that makes "additional assumptions".

There's a lot of material in section 2 which seems to be irrelevant: differences between classifiers and noun classes, age at which children exhibit competent use, semantic vs. arbitrary. All of this leaves the reader thinking that maybe some of these empirical facts will be things that the authors end up offering an explanation for, but as far as I can tell most of it plays no role. This includes the fact that the timecourse of a single child's acquisition demonstrates a move towards over-use of general classifiers; I don't understand how the simulations have any connection to that. (I'm still confused after lines 532-536.)

In section 3, I think I understand the important take-home message of the corpus study itself, but the presentation is somewhat confusing at times.
	- The logic at lines 244-248 seems slightly wrong. The preceding paragraph already established a candidate cause of greater homophony in Mandarin than Cantonese. Given this, if it's *also* true that homophony avoidance drives change, then we'd expect more classifier disambiguation in Mandarin than in Cantonese. The phrasing in the paper seems to suggest that the homophony avoidance hypothesis plays a part in predicting the increased ambiguity in Mandarin.
	- As far as I can tell, the terms "homophony" and "ambiguity" are used more or less interchangeably. This isn't necessarily incorrect, but the logic could be clearer if the authors stuck to just one term.
	- Similarly, the phrase "homophony avoidance" is somewhat confusing. As defined by the authors (line 266), the degree of homophony is just determined by the particular collection of form-transcription pairs that occur in the corpus. (It might also help to note that the transcription effectively determines the pronunciation.) So there's no way for classifier usage to "avoid" homophony. So something like "homophony mitigation" or "homophony circumvention" would be better than "homophony avoidance". (Perhaps the authors intended to distinguish between *homophony* and *ambiguity avoidance*, which would make sense, but the usage doesn't seem to be consistently in line with that.)
	- It might help to mention that matching on types means matching on *character* types, not *transcription* types.
	- In Table 1, why is the number of types for Mandarin_type not exactly 1182?

It's (mostly) unclear what the authors really want the reader to take away from the presentation of the results in section 4. They mention three findings (lines 456), but only one is clear the me: the one about moving towards generality, in the paragraph beginning at line 464. The second finding is something about the fact that the simulations tend to reach steady states (476-488), but I can't find any mention of what empirical point this is intended to align with. The last paragraph (489-501) seems to be framed as the third finding, but again this just seems to be a report of how certain simulations turned out, rather than an identifiable trend that meaningful aligns with an empirical target. A few lines later this is all just summarized as "provides insight into the long-term dynamics of classifier systems" (507), but the authors don't even seem to be trying to express what that insight is.

The opposition between "input" and "functional factors" (line 219) seems like a category error to me. Certainly there is always a tension between the input and generalization in any form of learning; but then the question of what role functional factors play is a question about the nature of the generalization that occurs, not the balance between input and generalization. I had a similar reaction around lines 506: would a proponent of functional pressures say that their account is not "acquisition-driven"? The idea that acquisition drives change (which is virtually a truism) seems orthogonal to the question about functional pressures.

---

### Decision · Program_Chairs · 2022-03-29

Reject